# The Influence of Agricultural Practices, the Environment, and Cultivar Differences on Soybean Seed Protein, Oil, Sugars, and Amino Acids

**DOI:** 10.3390/plants9030378

**Published:** 2020-03-19

**Authors:** Nacer Bellaloui, Angela M. McClure, Alemu Mengistu, Hamed K. Abbas

**Affiliations:** 1Agriculture Research Service, Crop Genetics Research Unit, USDA, P. O. Box 345, Stoneville, MS 38776, USA; 2West Tennessee Research & Education Center, Department of Plant Science, University of Tennessee, 605 Airways Boulevard, Jackson, TN 38301, USA; athomp15@utk.edu; 3Agricultural Research Service, Crop Genetics Research Unit, USDA, 605 Airways Boulevard, Jackson, TN 38301, USA; alemu.mengistu@ars.usda.gov; 4Agricultural Research Service, Biological Control of Pests Research Unit, USDA, P. O. Box 67, Stoneville, MS 38776, USA; hamed.abbas@ars.usda.gov

**Keywords:** cultivar differences, seed nutrition and composition, seed protein, seed sugars, seed oil, soybean, seed amino acids

## Abstract

Information on the effects of agricultural practices such as seeding rate (S), row spacing (RS), herbicide apical treatment (T), and nitrogen application (N) on soybean seed nutrition (protein, oil, fatty acids, sugars, and amino acids) is limited. Although seed composition (nutrition) constituents are genetically controlled, agricultural practices and environmental conditions significantly influence the amount and quality of seed nutrition. Therefore, the objective of this research was to understand the responses of these seed composition constituents to these practices, the environment, and cultivar differences. Two-field experiments were conducted, in 2015 and 2016, in Milan, TN, USA. The experiments were irrigated with four replications and included: two soybean cultivars, two seeding rates, three different row spacings, two N rates, and Cobra herbicide apical treatment. The results showed significant effects of S, RS, N, and T on some seed composition constituents, including protein; oleic, linolenic, and stearic acids; sugars; and some amino acids. The current research demonstrated that single or twin row with a seeding rate of 40,000 seeds ha^−1^ resulted in higher protein, oleic, some sugars, and some amino acids. However, a high seeding rate of 56,000 seeds ha^−1^ resulted in lower protein, oleic acid, some sugars, and some amino acids due to plant competition for soil nutrients. Herbicide apical application of Cobra1X resulted in higher linolenic acid and some amino acids. Application of nitrogen resulted in higher protein, linolenic, and some amino acids. This research is beneficial to the scientific communities, including breeders and physiologists through advancing knowledge on the interactions between cultivars and environment for seed nutritional quality selection, and to soybean producers through consideration of best agricultural management to maintain high seed nutritional qualities.

## 1. Introduction

Soybean is a major crop in the world due to its seed nutritional value as a food for human consumption and feed for livestock. Soybean seeds contain on a dry weight basis approximately 40% to 45% protein, 18% to 24% oil, 9% to 13% palmitic, 3% to 5% stearic, and 18% to 26% oleic acids, sugars, amino acids, isoflavones, and minerals (seed composition) [1,2,3,4,5]. Although seed composition is genetically controlled, it has been reported to be influenced by agricultural practices, including seeding rate, row spacing, irrigation, growing conditions, geographic location, and fertilizers. Research on the effects of agricultural practices on seed composition is limited [3,4] and results obtained are still conflicting [6,7]. Therefore, further investigation is needed before recommendations are made. The current research was aimed at investigating the effects of row spacing, seeding rate, and N fertilizer application on seed protein, oil, fatty acids, sugars, and amino acids.

In spite of the conflicting results, the effects of seeding rate, row spacing, and fertilizers have been previously reported [3,4,8,9,10,11,12]. For example, research on the effects of row spacing on yield showed that narrower row spacing increased the soybean yield, especially at late planting dates in the southeastern USA [12,13]. Wide row typically refers to an inter-row spacing of 100 cm; however, narrow row is 75 cm or less [14]. Determining the optimum row spacing and its effects on yield showed conflicting results [15]. It has been reported that narrower row spacing increased yield [13,16,17,18], but others [19,20,21] have reported that it did not. The explanation reported was that crop growth depended on the amount of radiation intercepted by the crop [22,23]. Therefore, yield increase was related to decreased distance between rows and increases in light interception at the critical phase for seed set [22,23]. In addition, the increase of yield has been attributed to minimum competition for water, minerals, and light, although other researchers have reported that little yield improvement was achieved when row spacing was less than 50 cm [24] or reduced yield [18]. It has been reported that yield increases were achieved by narrower row spacing as compared with conventional row spacing (76 to 100 cm) [15]. They found that, at a late planting date, soybean planted at row spacings of 100, 50, and 25 cm resulted in the highest yield at a row spacing of 50 cm. The yield at a row spacing of 25 cm was lower than at a row spacing of 50 cm, although light interception patterns were similar. They reported that radiation use efficiency during the seed filling stage was significantly lower at a row spacing of 25 cm as compared with row spacings at 50 and 100 cm.

The current research mainly focuses on the effects of row spacing and seeding rate on seed composition. Previous researches on the effects of row spacing and seeding rate on seed composition are limited, and those available are still conflicting. For example, the effects of row spacing on protein, oil, and fatty acid content were evaluated by [6]. They reported that oil content ranged from 20.9% to 22.3% and found that both oil and protein content were influenced by year and row spacing. They also found that a row spacing of 70 cm had the highest protein content followed by 60, 40, and 50 cm, respectively. Row spacing significantly affected oleic and linoleic acid contents [6]. Others have also evaluated the effects of seeding rate and row spacing in four soybean cultivars (P 93M90 and AG 3906, maturity group (MG) III; P 94B73, MG IV; and V 52N3, MG V and using seeding rates ranged from 247,000 to 592,800 seeds ha^−1^ for P 93M90 and AG 3906 and ranged from 60,000 to 180,000 seeds ha^−1^ for P 94B73 and V 52N3 [7]. Row spacings were 38 and 76 cm. They found that protein, oleic acid, sugars, P, and B concentrations increased with an increase of seeding rate in P 93M90 and AG 3906, but a decrease was observed at the highest seeding rate. This trend was reversed, in another year, due to environmental conditions of heat [7]. They concluded that seeding rate and row spacing altered seed composition, but these effects were dependent on the cultivar and the environmental factors, especially temperature and drought. They explained that the increase of protein concentrations with an increase of the seeding rates could be due to higher seeding rates that resulted in higher leaf area index, early season light interception, and early canopy closure [25], leading to an increase of nitrogen metabolism rates and higher seed protein accumulation. It has been reported that the genetic improvement in seed protein is associated with assimilate availability per seed and total assimilate supply per plant or unit area which could be associated with seed protein [2]. In addition, it has been reported that the decrease of seed protein concentration at higher seeding rates could be due to the interplant competition for soil nutrients availability [7] and soil moisture and shade effect at higher seeding rates [26]. This interpretation of interplant competition for soil nutrients was supported by a decrease in seed P, B, and Fe concentrations at the highest seeding rates. It has also been reported that the decrease in linolenic acid was due to the inverse relationship between linolenic and oleic acids [27,28,29,30,31,32]. Protein concentrations, in 2007, were lower than, in 2006, which could be due to higher temperatures and drought stress in 2007 [7]. The opposite trend occurred for protein and oleic acid concentrations with seeding rates which could be due to higher temperatures and drought stresses shown in some years [33]. The different responses of seed composition, including protein, oil, and fatty acids, to temperature [28,29,33] and drought have also been previously reported by [32,34,35,36,37].

Bellaloui et al. (2014) concluded that seeding rates and row spacing can alter seed protein, oil, and fatty acids, but these effects depended on the rate used, row spacing, genotype, and the yearly environmental conditions. Studies on the effects of row spacing and irrigation on total seed protein, oil, and fatty acid composition were conducted in sesame seed, and it was found that total oil content of sesame varied from 46.4% to 51.5% and the oil and protein contents were significantly different among treatments [38]. They also found that protein content was significantly affected by row spacing and irrigation, and a row spacing of 70 cm had the highest protein and oleic acid content followed by row spacing of 60, 50, and 40 cm, respectively, but the lowest oleic acid and highest linoleic content was observed at a row spacing of 40 cm, confirming the inverse relationship between oleic and linoleic acid. Seed oleic acid was also found to be affected by row spacing. They found that the interactions of row spacing were also found to be significant for oleic and linolenic acid [38]. The effect of row spacing and seeding rate on soybean protein and oil content was investigated using four different row spacings of 20 cm (narrow row), 50 cm (conventional), 20/80 cm (twin rows), and crossed lines (50 cm), and three seeding rates (150, 300, and 450 thousand viable seeds ha^−1^) [4]. They found inconsistent effects of higher population density on protein and oil content. Furthermore, a increased light index in narrow row culture has been shown to be related more to increased leaf area index, associated with greater total dry matter, than to increased light index per unit leaf area index resulting from more equidistant plant arrangement [18].

## 2. Results

### 2.1. ANOVA Analysis

ANOVA showed that Y (year), Cv (cultivar), and (S) seeding rate and (RS) row-spacing had significant effects on some seed composition, especially protein, oil, and oleic acid (Table 1). The year interaction with configuration, N rate, or apical application (T) were also significant for some seed components such as oleic acid (Table 1). The interaction between configuration and N rate, between seeding rate and N rate, between seeding rate and apical application, and between N rate and apical application were significant for protein. Generally, F values for Y, Cv, S, RS, and N rate are bigger as compared with the F values for other factors and their interactions. This pattern was generally followed by some sugars and some amino acids (Table 2, Table 3 and Table 4). Since Y significantly interacted for some seed composition constituents, including protein, oils, sugars, and amino acids, results were presented by year.

### 2.2. Seed Composition as Influenced by Different Row Configurations in Cultivar AG 4632 in 2015

Using twin row configuration showed that protein, linoleic acid, and sucrose contents were higher at the 100,000 seeds/ac (40K) rate than at the 140,000 seeds/ac (56K) rate (Table 5). However, stearic, oleic, and linolenic acid were higher at the 56K rate than at the 40K rate. Amino acids also responded differently, for example, AS, SER, GLY, TYR, LYS, and ARG were higher at the 56K rate than at the 40K rate. The configuration of 40 cm row spacing, on the one hand, showed that using different seeding rates of 40K and 56K showed that oil, oleic, and linoleic acids were highest at the 40K rate as compared with at the 56K rate. On the other hand, protein, palmitic, and linolenic acids were higher at higher seeding rates as compared with lower rates. Amino acids also responded differently. For example, AS, GLU, CYS, LEU, LYS, and TRY were higher at the 40K rate than at the 56K rate. The configuration of 30 row spacing showed that oil, and stearic and linoleic acids were higher at the 40K rate than at the 56K rate. Protein and stachyose were higher at the 56K rate than at the 40K rate. Amino acids were also altered by row spacing and seeding rate. For example, GLU and LYS were higher at the 40K rate than at the 56K rate. Amino acids THR, SER, GLY, ALA, VAL, ISO, and TYR were higher at the 56K rate than at the 40K rate (Table 5).

### 2.3. Seed Composition as Influenced by Different Row Configurations in Cultivar AG 4632 in 2016

Using the twin row configuration resulted in higher linolenic acid and stachyose content at the 40K rate than at the 56K rate, but oil and linoleic acid contents were higher at the 56K rate than at the 40K rate (Table 6). Amino acids, CYS and TRY, acid contents were the only two amino acids that responded to the seeding rate configuration “twin row” in this cultivar, in 2016, but ISO, LEU, and TYR were higher at the 56K rate than at the 40K rate. No changes were observed in the rest of amino acids. The configuration of 15 row spacing resulted in higher palmitic, stearic, and linolenic acids, and raffinose contents at the 40K rate than at the 56K rate (Table 6). The configuration of 15 row spacing also resulted in higher of some amino acids, including ASP, THR, SER, CYS, VAL, HIS, and ARG at the 40K rate as compared with at the 56K rate. Oil, oleic acid, and stachyose were higher at the 56K rate than at the 40K rate. The row spacing of 30 inches resulted in higher oil and oleic acid contents at the 40K rate than at the 56K rate, but higher in palmitic, stearic, linoleic, and linolenic acids, and stachyose content at the 56K rate as compared with the 40K rate. Amino acids ASP, PRO, CYS, PHE, LYS, and ARG showed a higher content at the 56K rate than at the 40K rate and no changes were recorded in other amino acids (Table 6).

### 2.4. Seed Composition as Influenced by Different Row Configurations in Cultivar P 47T36R in 2015

The configuration “twin row” resulted in higher sucrose and stachyose contents at the 40K rate than at the 56K rate (Table 7). No changes were observed in protein; oil; stearic, palmitic, oleic, linoleic, and linolenic acids; and sucrose or raffinose contents. Amino acids also responded differently, for example, GLY, ALA, VAL, and TRY were higher at the 40K rate than at the 56K rate. However, GLU, CYS, ISO, LEU, PHE, LYS, and ARG were higher at the 140K rate than at the 100K rate. The configuration of 15 row spacing and using different seeding rates of 40K and 56K, on the one hand, resulted in higher oil, oleic acid, and sucrose and stachyose content at the 40K rate as compared with the 56K rate. On the other hand, protein, linoleic acid, and raffinose contents were higher at the 56K rate as compared with the 40K rate. Amino acid ASP was higher at the 40K rate than at the 56K rate. Amino acids, CYS and TRY, were higher at the 56K rate than at the 40K rate and no changes were recorded on other amino acids at either the 40K rate or the 56K rate. The configuration of 30 row spacing showed that palmitic, stearic, linoleic and linolenic acids were higher at the 40K rate than at the 56 K rate. Amino acids ASP, GLU, PRO, ARG, and TRY were higher at the 40K rate than at the 56K rate. THR and MET amino acids were higher at the 56 K rate than at the 40K rate (Table 7).

### 2.5. Seed Composition as Influenced by Different Row Configurations in Cultivar P 47T36R in 2016

The configuration of row spacing 40 cm in twin row form resulted in higher oil and sucrose contents, but lower in protein and stachyose contents at the 40K rate than at the 56K rate (Table 8). There were no changes in palmitic, stearic, oleic, linoleic, and linolenic acids content, and in sucrose and raffinose contents. Protein and stachyose contents were higher at the 56K rate than at the 40K rate. In addition, amino acids SER, PRO, GLY, and ALA contents were higher at the 40K rate than at the 56K rate, but ISO and AG were higher at the 56K rate than at the 40K rate. No changes were observed in other amino acids. For the configuration of 15 row spacing, palmitic acid, sucrose, and stachyose contents were higher at the 40K rate than at the 56K rate. Amino acids GLU, PHE, HIS, and TRY were higher at the 40K rate than at the 56K rate. Amino acids ASP, THR, SER, PRO, GLY, ALA, VAL, ISO, LEU, TYR, and ARG were higher at the 40K rate than at the 56K rate. A row spacing of 76 cm resulted in higher protein content at the 56K rate than at the 40K rate, but there were no changes in oil, palmitic, stearic, oleic, linoleic, linolenic acids, and sugars contents. Amino acids such as THR, SER, GLY, ALA, VAL, LYS, and HIS either did not change or increased higher at the 40K rate than at the 56K rate such as MET; or increased higher at the 56K than at the 40K rate such as ASP, GLU, PRO, CYS, ISO, LEU, TYR, PHE, ARG, and TRY. (Table 8).

### 2.6. Seed Composition Constituents as Influenced by Apical and N Application

Effects of apical and nitrogen applications are shown in Figure 1A–D, Figure 2A–D, and Figure 3A–D). In 2015, apical application of Cobra1X resulted in a higher content of some seed composition constituents, including fatty acids stearic and linolenic, and some amino acids, including GLU, LYS, and ASP (Figure 1A,C and Figure 2A). The rest of the seed composition constituents either did not change such as oil and amino acids PHE and LYS, or decreased such as protein, palmitic acid, and amino acids ASP, SER, LEU, and ARG, and sucrose and stachyose. In 2016, the apical application resulted also in higher linolenic acid and amino acids PRO, GLY, and VAL, and stachyose (Figure 1B,D and Figure 2B). Some constituents decreased such as protein, linoleic acid, and amino acids ASP, GLU, and ARG, sucrose, and raffinose and no changes were noticed in some constituents, including palmitic and oleic acid acids, and the rest of amino acids.

In 2015, the application of nitrogen resulted in higher protein, oleic, and amino acids GLU, PHE, and LEU (Figure 2C and Figure 3C). However, other constituents showed a decrease with the application of N, including oil and sucrose (Figure 2C and Figure 3A) and no changes in the rest of amino acids ASP, SER, CYS, TYR, HIS, ARG, and TRY. In 2016, the application of N resulted in higher protein, linoleic and linolenic acids, and amino acids GLU, PHE, and ARG, and stachyose (Figure 2D and Figure 3B,D). Others decreased with the application of N such as oil, oleic acid, and sucrose. No changes were observed in stearic or palmitic acids, and amino acids SER, LEU, TYR, LYS, HIS, and TRY.

Generally, the cultivar AG 4632 accumulated higher protein and sucrose than in P 47T36R, in 2015 and 2016. As protein and oil contents are inversely correlated, oil was higher in P 47T36R than in AG 4632, in 2015 and 2016. Other fatty acids, sugars, and amino acids were accumulated at different rates and cultivars accumulated at different rates for each constituent due to cultivar differences.

## 3. Discussion

### 3.1. Effects of Row Spacing and Seeding Rate on Seed Protein, Oil, Fatty Acids, Sugars, and Amino Acids

Row spacing altered some seed composition constituents, and this alteration was dependent on row spacing, seeding rate, year, and cultivar. For example, on the one hand, in 2015, the twin row configuration resulted in higher protein, linolenic acid, and sucrose at a seeding rate of 40K than at the 56K rate, but the 15 or 30 row spacings resulted in higher oil, oleic acid, and linoleic acid at the 40K rate than at the 56K rate. On the other hand, in 2016, for the twin row configuration at the 56K rate, stearic, oleic, and linolenic acids were higher than at the 40K rate. Whereas, for the 15 or 30 row spacings, protein was higher at the 56K rate than at the 40K rate. Although there is limited information in the literature on the effects of row spacing and seeding rates on seed composition constituents, the available information on the effects of row spacing and seeding rate on the amount and type of seed composition constituents is still conflicting. For example, it has been reported that oil and protein content were influenced by year and row spacing, and a row spacing of 70 cm had the highest protein content followed by 60, 40, and 50 cm, respectively [6]. Additionally, they found that row spacing significantly affected oleic and linoleic acid contents. In addition, using seeding rates that ranged from 247,000 to 592,800 seeds ha^−1^ for cultivars P 93M90 and AG 3906 and ranged from 60,000 to 180,000 seeds ha^−1^ for cultivars P 94B73 and V 52N3 it has been found that protein, oleic acid, sugars, P, and B concentrations increased with an increase of seeding rate in P 93M90 and AG 3906, but a decrease was observed at the highest seeding rate [7]. They also found that the trends of increase and decrease were the opposite in another year due to environmental conditions, especially heat and drought [7]. They concluded that seeding rate and row spacing influenced seed composition, but this influence was dependent on cultivar and environmental condition factors, especially temperature and drought. They further explained that the alteration of seed composition and increase of protein content with increased seeding rates could be due to higher leaf area index, early season light interception, and early canopy closure [25], leading to an increase in the nitrogen metabolism rates and higher seed protein accumulation. On the one hand, the increase of seed protein was reported to be associated with assimilates availability per seed and total assimilate supply per plant or unit area [2], and the decrease of seed protein at higher seeding rates was reported to be associated with the interplant competition for soil nutrients availability and soil moisture and shade effect at higher seeding rates [7,26,39]. On the other hand, it was found that the application of a large amount of N resulted in a decrease in protein and an increase in oil [40].

The higher linolenic acid and stachyose content at the 40K rate than at the 56K rate; the higher oil and linoleic acid contents at the 56K rate than at the 40K rate, and the higher stearic, oleic, and linolenic acid at the 56K rate than at the 40K rate in 2016, shown in our studies, indicated different responses of seed oil, fatty acids, and sugars to seeding rate. This general pattern was also shown in 2015 for different configurations (twin row, 15 and 30 row spacings). The different response of amino acids to seeding rate, for example, the higher amino acids AS, SER, TYR, LYS, and ARG at the 56K rate than at the 40K rate also indicated the significant influence of seeding rate on amino acids. The different responses of amino acids were also noticed on different row spacings and in each year, indicating the influence of agricultural practices on the amount and profile of amino acids. The higher oil, oleic, and linoleic acids at the 40K rate than at the 56K rate with 15 row spacing, and the higher protein, palmitic, and linolenic acids at the 56K than at the 40K rate with 15 row spacing, for example in 2016, indicated the significant influence of seeding rate and row spacing on oil and fatty acids.

Previous researches have shown that the decrease of linolenic acid and increase of oleic acid, depending on seeding rate and row spacing, could be due to the inverse relationship between linolenic and oleic acids [27,28,29,30,31,32]. The response pattern, in 2015, of some seed composition constituents, including oil, fatty acids, sugars, and amino acids was different from that of 2016 and this could be due to environmental conditions under which soybeans grew, especially temperature and drought. Since the experiment was irrigated, temperature was more important than drought. The temperatures in 2016 (26.2 ℃ in June, 28 ℃ in July, 26.9 ℃ in August, 23.9 ℃ in September, and 16.8 ℃ in October) were higher than in 2015 (25.6 ℃ in June, 27.4 ℃ in July, 24.3 ℃ in August, 22.6 ℃ in September, and 15.9 ℃ in October). It was shown that protein concentrations, in 2007, were lower than, in 2006, which could be due to higher temperatures and drought stress in 2007, and the opposite trend of protein and oleic acid concentrations that showed in different years with seeding rates was explained to be due to higher temperature and drought stresses exhibited in some years [33]. Detailed reports on the effect of temperature on seed composition, including protein, oil, and fatty acids and their responses to environmental factors, including temperature and drought, have been previously shown in [28,29,33,41] and drought in [34,38,42,43].

Our results agree with the reports that seeding rates and row spacing alter seed composition, including changes in the amount and profile of oil (fatty acids), protein (amino acids), and sugars (sucrose, raffinose, and stachyose), and the pattern of changes could also depend on environmental conditions, including temperature, as the temperature during the critical stages of growth in 2016 was higher than in 2015, as shown above. Our results also showed inverse relationships between some seed composition constituents, for example, between protein and oil; and between oleic acid and linoleic or linolenic acid. Some seed constituents in cultivar AG 4632 showed a consistent pattern in two years, for example, higher oil, and higher oleic acid and linoleic acids with 30 row spacing at the 40K rate for two years, and an increase of sucrose and stachyose with 15 row spacing at the 40K rate for two years, which indicated the consistent responses of these constituents to this agricultural practice for this cultivar. Cultivar P 47T36R responds to row spacing and year were consistent in some seed constituents; however, other constituents showed variability, depending on year and row spacing. For example, stachyose was higher with twin row formation at the 40K rate as compared with the 56K rate in two years; higher sucrose and stachyose was shown with 15 row spacing at the 40K rate in two years; protein and raffinose increased with both 15 and 30 row spacing at the 56K rate for two years. Other constituents were not consistent in cultivar P 47T36R such as oil and fatty acids, where these constituents varied depending on row spacing, seeding rates, and year. The different responses of amino acids to row spacing, seeding rate, and environmental factors in each year were shown as an increase of some amino acids and a decrease in others, affecting amount and profile. The alterations of seed protein, oil, fatty acids, sugars, and amino acids with seeding rate and row spacing were explained in terms of radiation index, radiation interception, and plant-to-plant competition for available water, nutrients, and light. It has been reported that narrower spacing resulted in higher radiation index during the critical stage of seed set and the higher response for narrow rows was associated with increases of light interception during the seed set [23]. Reduced row spacing at equal plant densities resulted in equidistant plant distribution, decreased plant-to-plant competition for available water, nutrient, and light, and increased radiation interception and biomass production [44,45]. Narrow vs. wide row practices have also been also explained in terms of increased light index during critical developmental stages [18,44], and reduction of the row width from 100 to 75, and then to 50 cm [18] resulted in greater light index, higher growth rate, and increased production and development between emergence and seed initiation [14].

### 3.2. Effects of Apical Treatment and Nitrogen Application on Seed Protein, Oil, Fatty Acids, and Sugars

The increase of linolenic acid, in 2015 and 2016, by apical application indicated the positive influence of the apical application on this fatty acid. An increase of sucrose and decrease of stachyose, in 2015, and an increase of stachyose, in 2016, indicated the different responses of sugars to apical application. The decrease of different sugars in each year could also reflect the sensitivity of these sugars to the environmental condition factors, especially temperature, as the temperature in each year was different, especially during the critical period of growth stages (from June through September), as explained above. A similar response was shown for amino acids SER and LEU, in 2015, and Glu, in 2016, in that they showed different responses, probably due to temperature differences in each year. Similarly, for an increase of GLU, LYS, and ASP, in 2015, an increase of PRO, GLY, and VAL, in 2016.

The increase of protein and some amino acids, in 2015 and 2016, and oleic acid, in 2015, indicated the positive effects of N on seed protein, amino acids, and oleic acid. The decrease of oil and oleic acid, in 2016, could be due to the inherited inverse relationship between protein and oil and between oleic and linoleic or linolenic acids. The positive effects of growth regulators, biologically active substancess and growth stimulators on seed nutrition are limited, but they have been reported in a few studies. For example, in a four-year trial, it was found that soybean seed protein and oil production were higher in seeds treated with Lexin mixture as compared with untreated seed, due to the contribution of the biologically active substance, including hormones, to the overall growth and development [46]. Lexin is a biologically active substance containing a mixture of humic acids, fulvic acids, and auxins. Similar results were found when seeds were treated with brassinosteroid. It was explained that growth regulators (phytohormones) were associated with plant bioenergy and photosynthetic pigments, forming protein [47]. In addition, growth regulators, including biologically active substances, including synthetic auxins, humic acids, and fulvic acids had beneficial effects on seed germination and growth of soybean plants [46]. It was concluded that Lexin promotes faster cell division and growth, including tissues, vascular bundles, and whole plant health. It was reported that stimulatory effects of compounds such as biologically active, including humic substances, had positive effects on plant growth seed germination, root initiation, and total plant biomass, although there was a lack of understanding of the plant growth promotion mechanism [48]. In addition, bioactive substance has been reported to be associated with enhancing nutrients uptake, such as in case of Zn and Fe, promoting growth and development. The contribution of growth stimulators, including phytohormones, have been reported in detail in [49,50,51,52]. The apical application could have similar effects as biologically active substance.

The increase of protein, in 2015 and 2016, with N application; decrease of oil in 2015; decrease of sucrose and stachyose, in 2015, and sucrose and raffinose, in 2016; increase in amino acids such as GLU, PRO, ISO-LEU, and LEU, in 2015, and ASP, THR, GLU, GLU, and CYS, in 2016; decrease of linoleic and linolenic acids, in 2016; decrease of amino acids THR, GLY, and ALA, in 2015 and PRO, ALA, and MET, in 2016; and no change such as SER, LEU, TYR, and PHE, indicate the different seed composition constituents’ responses to N application. The effects of fertilizers, including N have been previously reported. For example, two separate field experiments, from 2005 to 2007, were conducted, where N was applied at 112 kg ha^−1^, and they found, on the one hand, a consistent increase in seed protein and oleic acid concentrations, and a decrease in oil and linolenic acid concentrations as compared with non-treated soybean [53]. They also found that the application of S + N increased the percentage up to 11.4% and 48.5% for protein and oleic acid, respectively. However, oil concentration decreased by 3%. In addition, they found that the increase of protein and oleic acid content was accompanied by a higher percentage of leaf and seed N and S. They also found that oil and linolenic acid concentrations were lower under drought conditions, indicating the significant effects of environmental condition factors on seed compositions. They concluded that S and N can alter seed composition under irrigated or non-irrigated conditions. Other studies have shown that higher percentages of protein and oleic acid were accompanied by higher concentrations of N, K, B, and Zn in seed [54]. Additionally, it was found that foliar boron application resulted in higher seed protein and oleic acid [28,30]. On the other hand, the effect of a large application of N on seed protein and oil was studied and found a significant decrease in seed protein concentration (1.9% for non-irrigated and 2.7% for irrigated), but an increase in oil concentration (2.2% for non-irrigated and 2.7% for irrigated) [40]. Although the essential role of N is well established, the effects of N on seed composition, including protein, oil, and fatty acids still needs further research before conclusive recommendations are made. It has been reported that protein content varies depending on the environmental stress factors such as temperature [29,55] and drought [55]. Previous research found that severe drought resulted in a decrease of protein content [38]; however, other studies showed the opposite trend that severe drought [28,55] increased protein concentration by 4.4% and 10.8%, respectively, and oil concentration and content decreased by 2.9% and 18.0%, respectively [32]. It was shown that protein and oil were negatively correlated (r = –0.87), indicating their inverse relationship [29,32,56]. It was explained that environmental stress factors such as temperature and drought during seed fill led to the deposition of a greater proportion of protein at the expense of oil, and 14.8% more protein and 18.3% less oil [32]. Our results with N applications consistently showed that the N applications resulted in higher protein, in 2015 and 2016, but a decrease in oil, in 2016, which were in agreement with previous findings [28,30,54], but in disagreement with other findings [40]. The increase of oleic acid under higher temperature suggests a possible role of this acid under environmental or chemical stresses. The inconsistent results reported in the literature of the effect of maturity and cultivar and environmental conditions, including temperature and drought on protein and fatty acids could be due to cultivar and maturity differences, and cultivar/genotype × environment interactions [29,34,57,58,59], although the effects of temperature on enzymes controlling fatty acids production were also reported, for example in [7].

## 4. Materials and Methods

### 4.1. Growth Conditions and Field Management

Field experiments were conducted, in 2015 and 2016, at the Milan Research and Education Center (MREC), Milan, TN, USA. The experiment was irrigated with 4 replications of the following treatments in a split-plot design: Two cultivars (Asgrow 4632 and Pioneer 47T36) (Jimmy Sanders Seed Inc., Jackson, TN); two planting densities of 100,000 seeds ac^−1^ (40,000 seeds ha^−1^) and 140,000 seeds ac^−1^ (56,000 seeds ha^−1^) (abbreviated as 100K and 140K, respectively); three row configurations (15 inch (40 cm) 2:1 twin row, 15 inch (40 cm) row spacing, and 30 inch (76 cm) row spacing); two N rates (0, 150 lbs N ac^−1^ and 0 or 168 kg ha^−1^) (abbreviated throughout the manuscript as 0 rate or 150 rate, respectively); Cobra herbicide (12 oz ac^−1^ = 0.88 L ha^−1^) applied at the growth stage V3 (the third trifoliolate stage with three trifoliolates unrolled) soybean on subset of plots (abbreviated to 0 and apical treatment, respectively). Cobra herbicide (lactofen is the active ingredient with the chemical name of 2-ethoxy-1-methyl-2-oxoethyl 5-[2-chloro-4-(trifluoromethyl)phenoxy]-2-nitrobenzoate]. Cobra herbicide was applied to control weeds and showed a suppression of the soybean disease white mold caused by *Sclerotinia sclerotiorum,* used to damage the apical meristem, thereby removing apical dominance and altering plant hormone levels.

All plots were planted with a 40 cm modified John Deere MaxEmerge no-till planter outfitted with a Rawson Accu-Rate (Rawson Control Systems, Inc., Oelwein, IA) controller. Twin rows were obtained by turning off every third planter unit and 76 cm rows were obtained by shutting off every other unit. Plots were planted on 6 May, 2015 and 6 May, 2016. Plots were irrigated using a center pivot system applying irrigation during each season according to the University of Tennessee MOIST irrigation scheduler (http://www.utcrops.com/irrigation/irr_mgmt_moist.htm) [60]. 2015 had high rainfall with moderate temperatures, while 2016 was a low rainfall year with hotter temperatures, during the mid-growing season. Ammonium nitrate (34-0-0) at 0 or 150 lbs N ac^−1^ (168 kg N ha^−1^) was applied by hand as a split application with half applied at the R1 growth stage and half applied at the R3 (beginning of pot set) growth stage. Stand counts were made at the V3 (the three nodes on the main stem beginning with the unifoliate node) growth stage and harvest by counting plants in 10 row ft and converting to plants ac. Leaf area index (LAI) was measured at R1 (LAI-2000, Li-Cor Biosciences, Lincoln, NE). Plant height and node number were measured at R1 and harvest, and pod measurements made at harvest and expressed as total pod (average main + branch) plant^−1^ and average branch pod plant^−1^. Plots were harvested at maturity and mature seeds were analyzed for seed protein, oil, fatty acids, sugars, and amino acids.

### 4.2. Seed Protein, Oil, Fatty Acids, Sugars, and Amino Acid Analyses

Mature seeds at R8 were analyzed for protein, oil, fatty acids, and sugars (sucrose, raffinose, and stachyose) contents using a Diode Array Feed Analyzer AD 7200 (Perten, Springfield, IL). Seed analyses were conducted according to others [7,28,30,57]. Briefly, mature dry seeds were ground by a Laboratory Mill 3600 (Perten, Springfield, IL) and a sample of 25 g of ground seed were analyzed. Calibration equations were initially developed by the University of Minnesota and upgraded by the Perten company using Perten’s Thermo Galactic Grams PLS IQ software. The calibration equations were established based on laboratory protocols according to AOAC methods [61,62]. Protein, oil, and sugars (sucrose, raffinose, and stachyose) were expressed on a dry matter basis [29,35,58,59]. Fatty acids (palmitic, stearic, oleic, linoleic, and linolenic) were expressed on a total oil basis.

### 4.3. Seed Amino Acids

Mature seeds at the R8 stage (maturity stage) were analyzed for amino acids content using near-infrared (NIR) reflectance diode array feed analyzer (Perten, Springfield, IL) according to others [28,29,63,64]. Individual amino acids alanine (ALN), cysteine (CYS), valine (VAL), methionine (MET), isoleucine (ISO-LEU), leucine (LEU), tyrosine (TYR), phenylalanine (PHE), lysine (LYS), histidine (HIS), arginine (ARG), tryptophan (TRY), asparagine (ASP), threonine (THR), serine (SER), glutamine (GLU), proline (PRO), and glycine (GLY), were analyzed. Briefly, mature, dry seeds sample of approximately 25 g from each plot was ground by a Laboratory Mill 3600 (Perten, Springfield, IL) according to others [65,66]. The calibration equation was initially developed by the Department of Agronomy and Plant Genetics, University of Minnesota, St Paul, MN, using Thermo Galactic Grams PLS IQ software developed by the Perten company (Perten, Springfield, IL), and then updated by Perten company. The quantification equation was established using laboratory protocols according to the Association of Official Analytical Chemists [67] and use of initial 8540 samples spectra, resulting in accurate estimations of amino acid quantification. Measurement of amino acids content (%) was based on dry matter.

### 4.4. Statistical analyses

Analysis of variance (ANOVA) was conducted using PROC MIXED (SAS, SAS Institute, 2002–2010). Replicate within year was considered as random effects. Seeding rate (S), row spacing (RS), herbicide apical treatment (T), nitrogen application (N), and cultivar were considered as fixed effects. Mean comparison procedures was conducted using SAS Macro PDMIX800 in PROC MIXED (SAS, SAS Institute, 2002–2010) [68] at significance level of P ≤ 0.05. Because the interactions involving combinations of year (Y), cultivar (Cv), configuration/row spacing (RS), seeding rate (S), nitrogen rate application (N), and apical application (T), were significant for some seed composition constituents, and results were presented separately by year and cultivar.

## 5. Conclusions

The current research demonstrated that agricultural practices, including seeding rate, row spacing, herbicide as apical application, and nitrogen application can alter seed composition constituents, including protein, oil, fatty acids, sugars, and some amino acids. For example, agricultural practices involving the use of single row or twin row with a seeding rate of 40,000 seeds ha^−1^ resulted in higher protein, oleic, some sugars, and some amino acids. However, a high rate of 56,000 seeds ha^−1^ resulted in lower protein, oleic acid, some sugars, and some amino acids due to plant competition for soil nutrients and soil moisture. Herbicide apical application of Cobra1X led to higher linolenic acid and some amino acids. Application of nitrogen resulted in higher protein, linolenic, and some amino acids. Therefore, these agricultural practices can alter some seed nutrients and we recommend using single row or twin-row practices with moderate seeding such as 40,000 seeds ha^−1^. Using a high seeding rate such as 56,000 seeds ha^−1^ can lead to interplant competition for soil nutrients, for seed quality. Therefore, considering best practices to maintain or increase specific nutrients to increase seed nutritional qualities is critical. Future research using developed drought- and heat-tolerant varieties with commercial cultivars under irrigated and non-irrigated conditions in multiyear field experiments should further advance our understanding of seed composition responses mechanisms to agriculture practices.

## Figures and Tables

**Figure 1 plants-09-00378-f001:**
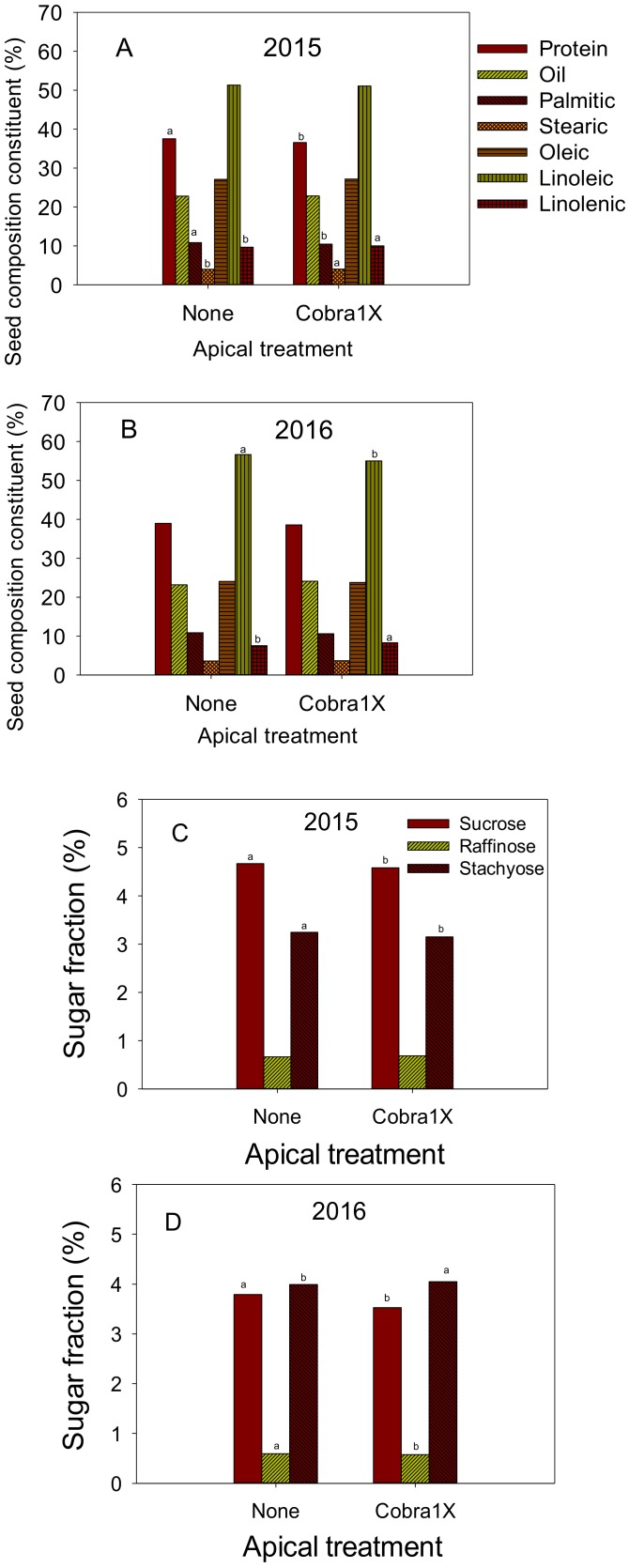
Effect of apical treatment of Cobra1X (rate of 12 oz/ac = 0.88 L ha^−1^) on seed protein, oil, and fatty acids palmitic, stearic, oleic, linoleic, and linolenic (%) in 2015 (**A**) and 2016 (**B**); Effect of apical treatment of Cobra1X (rate of 12 oz/ac = 0.88 L ha^−1^) on seed sugars (sucrose, raffinose, and stachyose) (%) in 2015 (**C**) and 2016 (**D**). Bars with the same colors and patterns with different letters are significantly different at P = 0.05. Bars without letters are not significant.

**Figure 2 plants-09-00378-f002:**
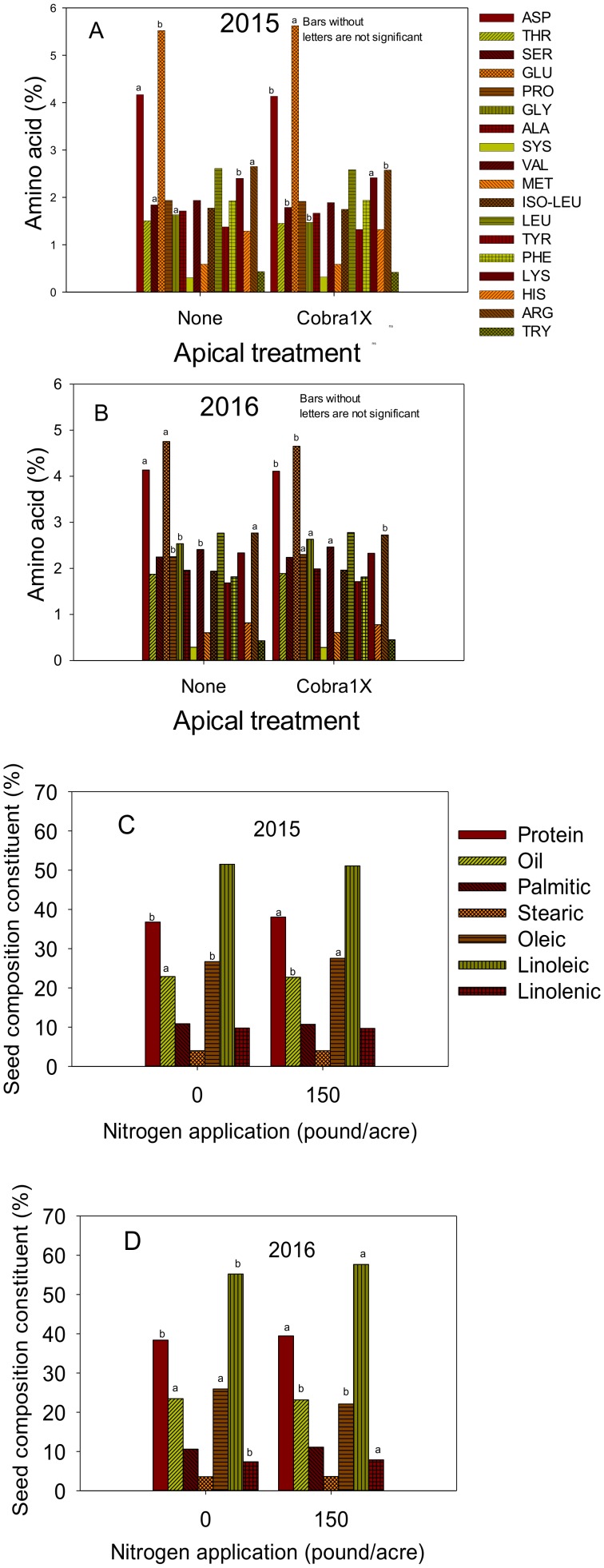
Effect of apical treatment of Cobra1X (rate of 12 oz/ac = 0.88 L ha^−1^) on seed amino acids (%) in 2015 (**A**) and 2016 (**B**); Effect of nitrogen (N) application at rate of 0 or 150 lbs N ac^−1^ (0 or 168 kg ha^−1^) on seed protein, oil, and fatty acids palmitic, stearic, oleic, linoleic, and linolenic (%) in 2015 (**C**) and 2016 (**D**). Bars with the same colors and patterns with different letters are significantly different at P ≤ 0.05. Bars without letters are not significant.

**Figure 3 plants-09-00378-f003:**
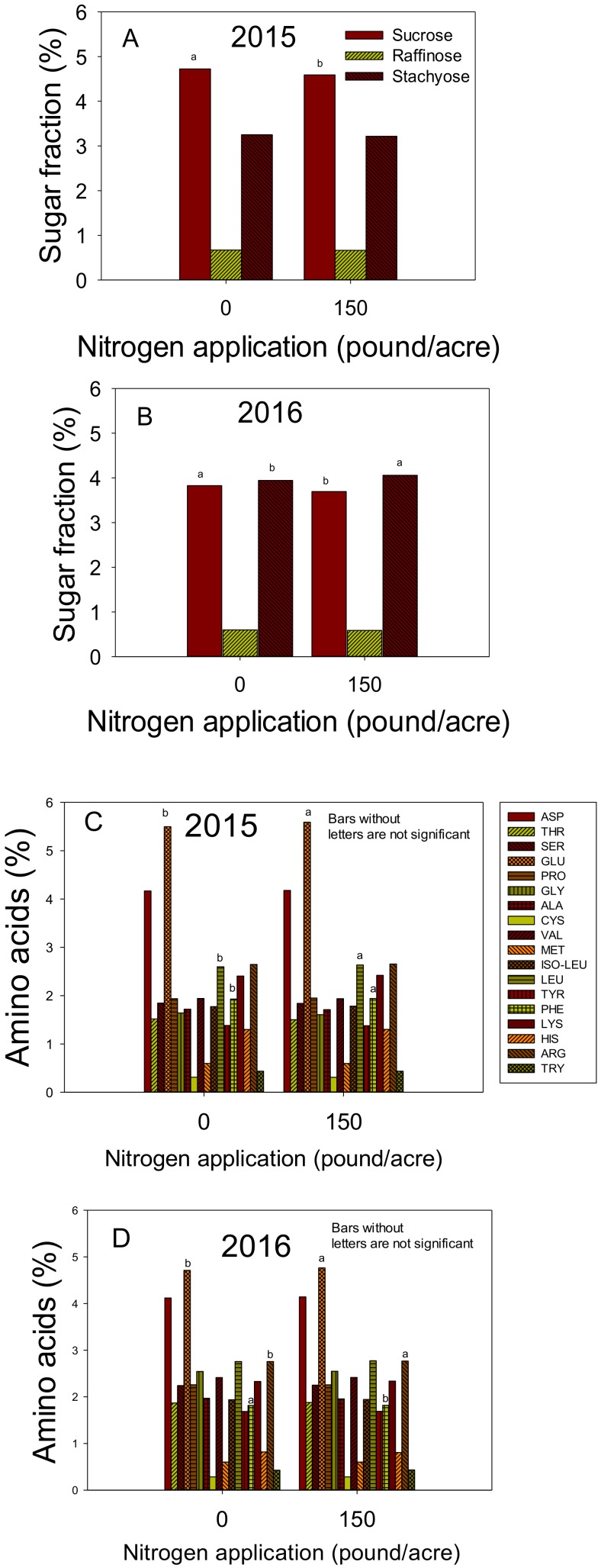
Effect of nitrogen (N) application at a rate of 0 or 150 lbs N ac^−1^ (0 or 168 kg ha^−1^) on seed sugars (sucrose, raffinose, and stachyose) (%) in 2015 (**A**) and 2016 (**B**); Effect of nitrogen (N) application at a rate of 0 or 150 lbs N ac^−1^ (0 or 168 kg ha^−1^) on seed amino acids (%) in 2015 (**C**) and 2016 (**D**). Bars with the same colors and patterns with different letters are significantly different at P ≤ 0.05. Bars without letters are not significant.

**Table 1 plants-09-00378-t001:** Analysis of variance (P values) for the effect of main effects of year (Y), cultivar (Cv), row spacing (RS), seeding rate (S), nitrogen rate (N), apical application (T), and their interactions on seed composition (protein, oil, palmitic (C16:0), stearic (C18:0), oleic (C18:1), linoleic (C18:2), and linolenic (C18:3 acids) (%). The experiments were conducted in 2015 and 2016.

	Protein	Oil	C16:0	C18:0	C18:1	C18:2	C18:3
Effect	P^e^	P	P	P	P	P	P
**Year (Y)**	***	*	ns	***	***	***	***
**Cultivar (Cv)**	***	***	*	ns	***	ns	ns
**Row spacing (RS)**	*	*	ns	ns	***	***	ns
**Seeding rate (S)**	ns^f^	ns	ns	ns	*	ns	ns
**N rate (N)**	***	ns	ns	ns	***	***	ns
**Apical application (T)**	ns	ns	ns	ns	ns	ns	ns
**Y*Cv**	ns	**	ns	ns	ns	*	ns
**Y* RS**	ns	ns	ns	ns	***	ns	ns
**Y*S**	ns	ns	ns	ns	ns	***	ns
**Y*N**	ns	ns	ns	ns	***	***	ns
**Y*T**	*	ns	ns	ns	ns	***	ns
**Cv * RS**	*	ns	ns	ns	***	ns	ns
**Cv * S**	ns	ns	ns	ns	ns	ns	ns
**Cv * N**	***	***	ns	ns	ns	ns	ns
**RS * S**	ns	ns	ns	ns	*	ns	ns
**RS * N**	**	ns	ns	ns	ns	***	ns
**S * N**	*	ns	ns	ns	ns	ns	ns
**S * T**	*	ns	ns	ns	***	ns	ns
**N * T**	*	ns	*	ns	***	*	*
**Y*Cv*RS*N*T**	*	ns	ns	ns	***	**	ns
**Residual**	0.90	0.83	1.9	0.10	5.27	3.86	2.70

Three row spacings were used as follows: 15 inches twin row; 15 inch row spacing (40 cm) which is abbreviated as 15 row spacing; 30 inch row spacing (76 cm) which is abbreviated as 30 row spacing. Seeding rate was used at 100,000 seeds/ac and 140,000 seeds/ac (i.e., 40,000 seeds ha^−1^ or 56,000 seeds ha^−1^ ) which is abbreviated as 40K and 56K throughout the manuscript, respectively. ^c^Nitrogen (N) was applied at a rate of 0 or 150 lbs N ac^−1^ (i.e., 0 or 168 kg ha^−1^). Apical treatment of Cobra1X (synthetic growth hormone stimulant) was applied at a rate of 12 oz/ac (i.e., 0.88 L ha^−1^). * Significance at P ≤ 0.05; ** significance at P ≤ 0.01; *** significance at P ≤ 0.001; ^f^ ns = not significant at P ≤ 0.05.

**Table 2 plants-09-00378-t002:** Analysis of variance (P values) for the effect of main effects of year (Y), cultivar (Cv), row spacing (RS), seeding rate (S), nitrogen rate (N), apical application (T), and their interactions on seed sugars (sucrose, raffinose, stachyose (%), and amino acids aspartic acid (ASP), threonine (THR), serine (SER), and glutamic (GLU) (%). The experiments were conducted in 2015 and 2016.

	Sucrose	Raffinose	Stachyose	ASP	THR	SER	GLU
Effect	P ^e^	P	P	P	P	P	P
**Year (Y)**	***	***	***	0.03	***	***	***
**Cultivar (Cv)**	***	***	ns	***	ns	ns	ns
**Row spacing (RS)**	ns^f^	ns	ns	***	ns	ns	*
**Seeding rate (S)**	ns	ns	ns	ns	ns	ns	ns
**N rate (N)**	ns	ns	ns	ns	ns	ns	ns
**Apical application (T)**	ns	ns	ns	ns	ns	ns	ns
**Y*Cv**	***	ns	ns	ns	ns	ns	*
**Y* RS**	ns	*	ns	ns	ns	ns	ns
**Y*S**	ns	ns	ns	ns	*	ns	*
**Y*N**	ns	ns	ns	ns	ns	ns	ns
**Y*T**	ns	ns	**	ns	**	ns	ns
**Cv * RS**	ns	ns	ns	ns	***	***	*
**Cv * S**	ns	ns	ns	ns	ns	ns	ns
**Cv * N**	ns	ns	ns	*	ns	ns	ns
**RS * S**	ns	ns	ns	ns	ns	ns	ns
**RS * N**	ns	ns	ns	ns	ns	ns	ns
**S * N**	ns	ns	ns	ns	ns	ns	ns
**S * T**	ns	ns	ns	ns	ns	ns	ns
**N * T**	ns	ns	ns	ns	ns	ns	ns
**Y*Cv*RS*N*T**	ns	**	ns	ns	**	ns	ns
**Residual**	0.1142	0.0017	0.07	0.0061	0.004	0.0130	0.0994

Three row spacings were used as follows: 15 inches twin row; 15 inch row spacing (40 cm) which is abbreviated as 15 row spacing; and 30 inch row spacing (76 cm) which is abbreviated as 30 row spacing. Seeding rate was used at 100,000 seeds/ac and 140,000 seeds/ac (i.e., 40,000 seeds ha^−1^ or 56,000 seeds ha^−1^ ) which is abbreviated as 40K and 56K throughout the manuscript, respectively. ^c^Nitrogen (N) was applied at a rate of 0 or 150 lbs N ac^−1^ (i.e., 0 or 168 kg ha^−1^). Apical treatment of Cobra1X (synthetic growth hormone stimulant) was applied at a rate of 12 oz/ac ( i.e., 0.88 L ha^−1^). * Significance at P ≤ 0.05; ** significance at P ≤ 0.01; *** significance at P ≤ 0.001; ^f^ ns = not significant at P ≤ 0.05.

**Table 3 plants-09-00378-t003:** Analysis of variance (P values) for the effect of main effects of year (Y), cultivar (Cv), row spacing (RS), seeding rate (S), nitrogen rate (N), apical application (T), and their interactions on seed amino acids proline (PRO), glycine (GLY), alanine (ALA), cystine (CYS), valine (VAL), methionine (MET), and iso-leucine (ISO) (%). The experiments were conducted in 2015 and 2016.

	PRO	GLY	ALA	CYS	VAL	MET	ISO
Effect	P ^e^	P	P	P	P	P	P
**Year (Y)**	***	***	***	*	***	*	***
**Cultivar (Cv)**	*	ns	ns	ns	ns	ns	ns
**Row spacing (RS)**	ns^f^	ns	ns	ns	ns	ns	*
**Seeding rate (S)**	ns	ns	ns	ns	ns	ns	ns
**N rate (N)**	ns	ns	ns	ns	ns	ns	ns
**Apical application (T)**	ns	ns	ns	ns	ns	ns	ns
**Y*Cv**	ns	ns	ns	ns	ns	ns	***
**Y* RS**	ns	ns	ns	ns	ns	ns	*
**Y*S**	ns	ns	ns	ns	ns	ns	ns
**Y*N**	ns	ns	ns	ns	ns	ns	ns
**Y*T**	*	***	***	ns	*	ns	ns
**Cv * RS**	***	***	*	ns	***	ns	***
**Cv * S**	ns	ns	ns	ns	ns	ns	ns
**Cv * N**	ns	ns	ns	ns	ns	ns	ns
**RS * S**	ns	ns	ns	ns	ns	ns	ns
**RS * N**	ns	ns	ns	ns	ns	ns	ns
**S * N**	ns	*	ns	ns	*	ns	ns
**S * T**	ns	ns	ns	ns	ns	ns	ns
**N * T**	ns	ns	ns	ns	ns	ns	ns
**Y*Cv*RS*N*T**	ns	*	*	ns	*	ns	ns
**Residual**	0.00	0.03	0.00	0.00	0.01	0.00	0.00

Three row spacings were used as follows: 15 inches twin row; 15 inch row spacing (40 cm) which is abbreviated as 15 row spacing; 30 inch row spacing (76 cm) which is abbreviated as 30 row spacing. Seeding rate was used at 100,000 seeds/ac and 140,000 seeds/ac (i.e., 40,000 seeds ha^−1^ or 56,000 seeds ha^−1^ ) which is abbreviated as 40K and 56K throughout the manuscript, respectively. Nitrogen (N) was applied at a rate of 0 or 150 lbs N ac^−1^ (i.e., 0 or 168 kg ha^−1^). Apical treatment of Cobra1X (synthetic growth hormone stimulant) was applied at a rate of 12 oz/ac (i.e., 0.88 L ha^−1^). * Significance at P ≤ 0.05; *** significance at P ≤ 0.001; ^f^ ns = not significant at P ≤ 0.05.

**Table 4 plants-09-00378-t004:** Analysis of variance (P values) for the effect of main effects of year (Y), cultivar (Cv), row spacing (RS), seeding rate (S), nitrogen rate (N), apical application (T), and their interactions on seed amino acids leucine (LEU), tyrosine (TYR), phenylalanine (PHE), lysine (LYC), histidine (HIS), arginine (ARG), and tryptophan (TRY) (%). The experiments were conducted in 2015 and 2016.

	LEU	TYR	PHE	LYS	HIS	ARG	TRY
Effect	P ^e^	P	P	P	P	P	P
**Year (Y)**	***	***	***	***	***	***	ns
**Cultivar (Cv)**	ns^f^	ns	ns	***	ns	***	ns
**Row spacing (RS)**	*	ns	ns	*	*	***	ns
**Seeding rate (S)**	ns	ns	ns	ns	ns	ns	ns
**N rate (N)**	ns	ns	ns	ns	ns	ns	ns
**Apical application (T)**	ns	ns	ns	ns	ns	*	ns
**Y*Cv**	*	ns	ns	ns	*	***	ns
**Y* RS**	ns	ns	ns	ns	ns	ns	ns
**Y*S**	*	ns	ns	ns	ns	*	ns
**Y*N**	ns	ns	ns	ns	ns	ns	ns
**Y*T**	ns	*	ns	ns	ns	ns	ns
**Cv * RS**	***	ns	ns	*	*	ns	ns
**Cv * S**	ns	ns	ns	ns	ns	ns	ns
**Cv * N**	ns	ns	ns	ns	ns	*	ns
**RS * S**	ns	ns	ns	ns	ns	ns	ns
**RS * N**	ns	ns	ns	ns	ns	ns	ns
**S * N**	ns	ns	ns	ns	ns	ns	ns
**S * T**	ns	ns	ns	ns	ns	ns	ns
**N * T**	ns	ns	ns	ns	ns	ns	*
**Y*Cv*RS*N*T**	ns	ns	ns	ns	ns	ns	ns
**Residual**	0.02	0.00	0.00	0.01	0.02	0.00	0.00

Three row spacings were used as follows: 15 inches twin row; 15 inch row spacing (40 cm) which is abbreviated as 15 row spacing; 30 inch row spacing (76 cm) which is abbreviated as 30 row spacing. Seeding rate was used at 100,000 seeds/ac and 140,000 seeds/ac (i.e., 40,000 seeds ha^−1^ or 56,000 seeds ha^−1^ ) which is abbreviated as 40K and 56K throughout the manuscript, respectively. Nitrogen (N) was applied at a rate of 0 or 150 lbs N ac^−1^ (i.e., 0 or 168 kg ha^−1^). Apical treatment of Cobra1X (synthetic growth hormone stimulant) was applied at a rate of 12 oz/ac (i.e., 0.88 L ha^−1^). * Significance at P ≤ 0.05; *** significance at P ≤ 0.001; ns = not significant at P ≤ 0.05.

**Table 5 plants-09-00378-t005:** The effect of seeding rate^a^, row spacing (15 inches twin row, 15 row spacing, and 30 row spacing (40 cm and 76 cm, respectively) on seed composition (protein; oil, palmitic (C16:0), stearic (C18:0), oleic (C18:1), linoleic (C18:2), linolenic (C18:3); sugars (sucrose, raffinose, stachyose) (%); and amino acids aspartic acid (ASP), threonine (THR), serine (SER), and glutamic acid (GLU), proline (PRO), glycine (GLY), alanine (ALA), cystine (CYS), valine (VAL), methionine (MET), iso-leucine (ISO), leucine (LEU), tyrosine (TYR), phenylalanine (PHE), lysine (LYC), histidine (HIS), arginine (ARG), and tryptophan (TRY) (%) in cultivar AG 4632 in 2015.

				**Twin row 40 cm**						
	**Protein**	**Oil**	**C16:0**	**C18:0**	**C18:1**	**C18:2**	**C18:3**	**Sucrose**	**Raffinose**	**Stachyose**
**40K**	38.8a	22.05a	10.09a	3.9b	27.8b	51.1a	9.4b	4.88a	0.69a	3.11a
**56K**	37.95b	22.28a	11.1a	4.1a	29.2a	50.1b	10.0a	4.69b	0.68a	3.06a
	**ASP**	**THR**	**SER**	**GLU**	**PRO**	**GLY**	**ALA**	**CYST**	**VAL**	
**40K**	4.23b	1.54a	1.86b	5.63a	1.99a	1.66b	1.74a	0.33a	1.96a	
**56K**	4.28a	1.54a	1.90a	5.71a	1.99a	1.73a	1.74a	0.33a	1.96a	
	**MET**	**ISO**	**LEC**	**TYR**	**PHE**	**LYS**	**HIST**	**ARG**	**TRY**	
**40K**	0.6a	1.76a	2.63a	1.38b	1.96a	2.43b	1.39a	2.70b	0.45a	
**56K**	0.6a	1.76a	2.65a	1.40a	1.97a		1.39a	2.73a	0.44a	
				**Row spacing 40 cm**						
	**Protein**	**Oil**	**C16:0**	**C18:0**	**C18:1**	**C18:2**	**C18:3**	**Sucrose**	**Raffinose**	**Stachyose**
**40K**	37.3b	22.9a	10.3b	4.06	23.3a	54.8a	8.1b	4.79a	0.686a	3.2a
**56K**	38.2a	21.9b	11.2a	4.09	24.0b	52.6b	9.6a	4.86a	0.675a	3.2a
	**ASP**	**THR**	**SER**	**GLU**	**PRO**	**GLY**	**ALA**	**CYS**	**VAL**	
**40K**	4.26a	1.51b	1.84b	5.71a	1.97a	1.61b	1.71b	0.34a	1.96b	
**56K**	4.19b	1.56a	1.93a	5.43b	1.99a	1.83a	1.77a	0.30b	2.01a	
	**MET**	**ISO**	**LEC**	**TYR**	**PHE**	**LYS**	**HIS**	**ARG**	**TRY**	
**40K**	0.59b	1.79a	2.69a	1.39a	1.96a	2.49a	1.29a	2.74a	0.443a	
**56K**	0.60a	1.80a	2.60b	1.41a	1.93a	2.38b	1.30a	2.73a	0.425b	
				**Row spacing 76 cm**						
	**Protein**	**Oil**	**C16:0**	**C18:0**	**C18:1**	**C18:2**	**C18:3**	**Sucrose**	**Raffinose**	**Stachyose**
**40K**	38.3b	22.84a	11.0a	4.05a	28.74a	51.15a	9.21b	4.68a	0.68a	3.13b
**56K**	39.0a	21.99b	11.2a	3.91b	28.60a	49.71b	10.51a	4.71a	0.68a	3.20a
	**AS**	**THR**	**SER**	**GLU**	**PRO**	**GLY**	**ALA**	**CYS**	**VAL**	
**40K**	4.24a	1.49b	1.79b	5.78a	1.94a	1.56b	1.69b	0.31a	1.89b	
**56K**	4.25a	1.54a	1.90a	5.55b	1.95a	1.80a	1.76a	0.30a	1.98a	
	**MET**	**ISO**	**LEC**	**TYR**	**PHE**	**LYS**	**HIS**	**ARG**	**TRY**	
**40K**	0.60a	1.80b	2.71a	1.36b	1.95a	2.49a	1.25a	2.73a	0.45a	
**56K**	0.60a	1.83a	2.69a	1.44a	1.94a	2.44b	1.23a	2.76a	0.46a	

^a^ Seeding rate was used at 100,000 seeds/ac and 140,000 seeds/ac (i.e., 40,000 seeds ha^−1^ or 56,000 seeds ha^−1^ ) which is abbreviated as 40K and 56K throughout the manuscript, respectively. ^b^ Three configuration forms were used as follows: 15 inches twin row; 15 inch row spacing (40 cm) which is abbreviated to 15 row spacing; and 30 inch row spacing (76 cm) which is abbreviated as 30 row spacing. Values are means of four replications. Values followed by different lowercase letters within a column within each treatment are different at P ≤ 0.05.

**Table 6 plants-09-00378-t006:** The effect of seeding rate and row spacing (15 inches twin row, 15 row spacing, and 30 row spacing (40 cm and 76 cm, respectively) on seed composition (protein; oil, palmitic (C16:0); stearic (C18:0); oleic (C18:1); linoleic (C18:2); linolenic (C18:3); sugars (sucrose, raffinose, stachyose) (%); and amino acids aspartic acid (ASP), threonine (THR), serine (SER), glutamic acid (GLU), proline (PRO), glycine (GLY), alanine (ALA), cystine (CYS), valine (VAL), methionine (MET), iso-leucine (ISO), leucine (LEU), tyrosine (TYR), phenylalanine (PHE), lysine (LYC), histidine (HIS), arginine (ARG), and tryptophan (TRY) (%) in cultivar AG 4632, in 2016.

			**Twin row 40 cm**							
	**Protein**	**Oil**	**C16:0**	**C18:0**	**C18:1**	**C18:2**	**C18:3**	**Sucrose**	**Raffinose**	**Stachyose**
**40K**	39.3a	22.9b	10.5a	3.46a	24.5a	55.6b	7.14a	4.09a	0.6a	4.04a
**56K**	39.6a	23.3a	10.7a	3.39a	25.3a	57.1a	6.26b	4.11a	0.6a	3.78b
	**ASP**	**THR**	**SER**	**GLU**	**PRO**	**GLY**	**ALA**	**CYS**	**VAL**	
**40K**	4.15a	1.91a	2.31a	4.79a	2.28a	2.63a	1.99a	0.31a	2.45a	
**56K**	4.18a	1.91a	2.29a	4.71a	2.28a	2.64a	1.99a	0.29b	2.46a	
	**MET**	**ISO**	**LEC**	**TYR**	**PHE**	**LYS**	**HIS**	**ARG**	**TRY**	
**40K**	0.60a	1.93b	2.71b	1.68b	1.83a	2.33a	0.85a	2.83a	0.43a	
**56K**	0.80a	1.95a	2.79a	1.74a	1.83a	2.34a	0.83a	2.85a	0.40b	
			**Row spacing 40 cm**	**2016**						
	**Protein**	**Oil**	**C16:0**	**C18:0**	**C18:1**	**C18:2**	**C18:3**	**Sucrose**	**Raffinose**	**Stachyose**
**40K**	39.6a	21.9b	11.9a	3.70a	21.5b	56.9a	7.91a	4.34a	0.64a	4.11b
**56K**	39.4a	22.9a	10.7b	3.55b	24.1a	57.4a	7.24b	4.23a	0.61b	3.93a
	**AS**	**THR**	**SER**	**GLU**	**PRO**	**GLY**	**ALA**	**CYST**	**VAL**	
**40K**	4.14a	1.94a	2.35a	4.53a	2.30a	2.64a	1.99a	0.263a	2.45a	
**56K**	4.09b	1.88b	2.30b	4.54a	2.28a	2.59a	1.99a	0.288b	2.41b	
	**MET**	**ISO**	**LEC**	**TYR**	**PHE**	**LYS**	**HIS**	**ARG**	**TRY**	
**40K**	0.60a	1.86a	2.64a	1.69a	1.81a	2.31a	0.98a	2.78a	0.415a	
**56K**	0.60a	1.88a	2.66a	1.69a	1.81a	2.29a	0.91b	2.75b	0.412a	
			**Row spacing 76 cm**	**2016**						
	**Protein**	**Oil**	**C16:0**	**C18:0**	**C18:1**	**C18:2**	**C18:3**	**Sucrose**	**Raffinose**	**Stachyose**
**40K**	39.9a	22.4a	10.95b	3.59b	25.5a	55.2b	7.36b	4.15a	0.613a	3.81b
**56K**	40.1a	21.5b	12.29a	3.76a	21.1b	57.6a	8.59a	4.09a	0.90a	3.94a
	**ASP**	**THR**	**SER**	**GLU**	**PRO**	**GLY**	**ALA**	**CYS**	**VAL**	
**40K**	4.13b	1.83a	2.19a	4.53a	2.21b	2.45a	1.94a	0.26b	2.38a	
**56K**	4.19a	1.83a	2.19a	4.54a	2.24a	2.43a	1.94a	0.30a	2.36a	
	**MET**	**ISO**	**LEU**	**TYR**	**PHE**	**LYS**	**HIS**	**ARG**	**TRY**	
**40K**	0.6a	1.96a	2.83a	1.68a	1.80b	2.38b	0.76a	2.80b	0.41a	
**56K**	0.6a	1.96a	2.86a	1.65a	1.86a	2.45a	0.80a	2.85a	0.41a	

^a^ Seeding rate was used at 100,000 seeds/ac and 140,000 seeds/ac (i.e., 40,000 seeds ha^−1^ or 56,000 seeds ha^−1^ ) which is abbreviated as 40K and 56K throughout the manuscript, respectively. ^b^ Three configuration forms were used as follows: 15 inches twin row; 15 inch row spacing (40 cm) which is abbreviated to 15 row spacing; 30 inch row spacing (76 cm) which is abbreviated as 30 row spacing)]. Values are means of four replications. Values followed by different lowercase letters within a column within each treatment are different at P ≤ 0.05.

**Table 7 plants-09-00378-t007:** The effect of seeding rate and row spacing (15 inches twin row, 15 row spacing, and 30 row spacing (40 cm and 76 cm, respectively) on seed composition (protein; oil, palmitic (C16:0), stearic (C18:0), oleic (C18:1), linoleic (C18:2), linolenic (C18:3); sugars (sucrose, raffinose, stachyose) (%); and amino acids aspartic acid (ASP), threonine (THR), serine (SER), glutamic acid (GLU), proline (PRO), glycine (GLY), alanine (ALA), cystine (CYS), valine (VAL), methionine (MET), iso-leucine (ISO), leucine (LEU), tyrosine (TYR), phenylalanine (PHE), lysine (LYC), histidine (HIS), arginine (ARG), and tryptophan (TRY) (%) in cultivar P 47T36R, in 2015.

				**Twin row 40 cm**						
	**Protein**	**Oil**	**C16:0**	**C18:0**	**C18:1**	**C18:2**	**C18:3**	**Sucrose**	**Raffinose**	**Stachyose**
**40K**	36.5a	23.3a	10.6a	4.0a	27.6a	51.5a	9.5a	4.73a	0.65a	3.45a
**56K**	36.4a	23.2a	11.0a	4.0a	27.0a	52.0a	9.4a	4.58b	0.65a	3.3b
	**ASP**	**THR**	**SER**	**GLU**	**PRO**	**GLY**	**ALA**	**CYS**	**VAL**	
**40K**	4.1a	1.49a	1.8a	5.3b	1.9a	1.6a	1.70a	0.30b	1.93a	
**56K**	4.1a	1.48a	1.8a	5.5a	1.9a	1.5b	1.68b	0.33a	1.89b	
	**MET**	**ISO**	**LEC**	**TYR**	**PHE**	**LYS**	**HIS**	**ARG**	**TRY**	
**40K**	0.6a	1.75b	2.55b	1.36a	1.91b	2.36b	1.30a	2.56b	0.44a	
**56K**	0.6a	1.79a	2.64a	1.38a	1.94a	2.43a	1.26a	2.61a	0.41b	
				**Row spacing 40 cm**						
	**Protein**	**Oil**	**C16:0**	**C18:0**	**C18:1**	**C18:2**	**C18:3**	**Sucrose**	**Raffinose**	**Stachyose**
**40K**	36.73b	23.75a	10.83a	3.94a	27.68a	51.4b	9.94a	4.58a	0.63b	3.29a
**56K**	37.24a	23.21b	11.01a	3.96a	26.41b	51.9a	10.30a	4.44b	0.66a	3.16b
	**ASP**	**THR**	**SER**	**GLU**	**PRO**	**GLY**	**ALA**	**CYST**	**VAL**	
**40K**	4.14a	1.49a	1.83a	5.41a	1.93a	1.59a	1.73a	0.31b	1.95a	
**56K**	4.08b	1.51a	1.80a	5.45a	1.91a	1.56a	1.73a	0.34a	1.91a	
	**MET**	**ISO**	**LEU**	**TYR**	**PHE**	**LYS**	**HIS**	**ARG**	**TRY**	
**40K**	0.60a	1.79a	2.63a	1.39a	1.91a	2.36a	1.26a	2.6a	0.425b	
**56K**	0.59a	1.79a	2.59a	1.39a	1.93a	2.39a	1.26a	2.6a	0.450a	
				**Row spacing 76 cm**						
	**Protein**	**Oil**	**C16:0**	**C18:0**	**C18:1**	**C18:2**	**C18:3**	**Sucrose**	**Raffinose**	**Stachyose**
**40K**	37.16a	22.87a	10.83a	4.06a	26.73b	51.40a	10.11a	4.6a	0.68a	3.24a
**56K**	36.95a	22.98a	10.37b	3.99b	27.7a	50.28b	9.80b	4.6a	0.67b	3.3b
	**ASP**	**THR**	**SER**	**GLU**	**PRO**	**GLY**	**CYS**	**ALA**	**VAL**	
**40K**	4.17a	1.49b	1.83a	5.60a	1.95a	1.57a	0.308a	1.70a	1.94a	
**56K**	4.14b	1.51a	1.84a	5.47b	1.93b	1.60a	0.317a	1.71a	1.93a	
	**MET**	**ISO**	**LEC**	**TYRO**	**PHE**	**LYS**	**HIS**	**ARG**	**TRY**	
**40K**	0.595b	1.771a	2.596a	1.37a	1.94a	2.390a	1.33a	2.621a	0.438a	
**56K**	0..600a	1.763a	2.579a	1.36a	1.93a	2.395a	1.30a	2.595b	0.425b	

^a^ Seeding rate was used at 100,000 seeds/ac and 140,000 seeds/ac (i.e., 40,000 seeds ha^−1^ or 56,000 seeds ha^−1^ ) which is abbreviated as 40K and 56K throughout the manuscript, respectively. ^b^ Three configuration forms were used as follows: 15 inches twin row; 15 inch row spacing (40 cm) which is abbreviated to 15 row spacing; 30 inch row spacing (76 cm), which is abbreviated as 30 row spacing. Values are means of four replications. Values followed by different lowercase letters within a column within each treatment are different at P ≤ 0.05.

**Table 8 plants-09-00378-t008:** The effect of seeding rate and row spacing (15 inches twin row, 15 row spacing, and 30 row spacing (40 cm and 76 cm, respectively) on seed composition (protein; oil, palmitic (C16:0), stearic (C18:0), oleic (C18:1), linoleic (C18:2), linolenic (C18:3); sugars (sucrose, raffinose, stachyose) (%); and amino acids aspartic acid (ASP), threonine (THR), serine (SER), and glutamic acid (GLU), proline (PRO), glycine (GLY), alanine (ALA), cystine (CYS), valine (VAL), methionine (MET), iso-leucine (ISO), leucine (LEU), tyrosine (TYR), phenylalanine (PHE), lysine (LYC), histidine (HIS), arginine (ARG), and tryptophan (TRY) (%) in cultivar P 47T36R, in 2016.

				**Twin row 40 cm**						
	**Protein**	**Oil**	**C16:0**	**C18:0**	**C18:1**	**C18:2**	**C18:3**	**Sucrose**	**Raffinose**	**Stachyose**
**40K**	38.2b	24.14a	11.10a	3.61a	24.2a	55.73a	7.6a	3.65a	0.588a	4.06b
**56K**	39.0a	22.68b	11.13a	3.69a	24.1a	55.56a	7.9a	3.41b	0.575a	4.15a
	**ASP**	**THR**	**SER**	**GLU**	**PRO**	**GLY**	**ALA**	**CYS**	**VAL**	
**40K**	4.13a	1.85a	2.24a	4.86a	2.21a	2.48a	1.95a	0.30a	2.36a	
**56K**	4.14a	1.84a	2.20b	4.83a	2.18b	2.41b	1.91b	0.30a	2.36a	
	**MET**	**ISO**	**LEC**	**TYR**	**PHE**	**LYS**	**HIS**	**ARG**	**TRY**	
**40K**	0.6a	1.91b	2.76a	1.65a	1.84a	2.33a	2.73a	2.80b	0.44a	
**56K**	0.6a	1.96a	2.79a	1.66a	1.83a	2.34a	2.73a	2.85a	0.44a	
				**Row spacing 40 cm**						
	**Protein**	**Oil**	**C16:0**	**C18:0**	**C18:1**	**C18:2**	**C18:3**	**Sucrose**	**Raffinose**	**Stachyose**
**40K**	38.3b	23.9a	10.99a	3.51a	23.9a	57.88b	7.79a	3.64a	0.6a	3.90a
**56K**	38.9a	24.1a	10.25b	3.50a	24.6a	59.9a	7.71a	3.38b	0.6a	4.08b
	**ASP**	**THR**	**SER**	**GLU**	**PRO**	**GLY**	**ALA**	**CYS**	**VAL**	
**40K**	4.09b	1.80b	2.16b	4.85a	2.19b	2.39b	1.91b	0.275a	2.33b	
**56K**	4.11a	1.85a	2.20a	4.73b	2.23a	2.50a	1.94a	0.275a	2.40a	
	**MET**	**ISO**	**LEC**	**TYR**	**PHE**	**LYS**	**HIST**	**ARG**	**TRY**	
**40K**	0.60a	1.93b	2.76b	1.65b	1.81a	2.34a	0.81a	2.70b	0.43a	
**56K**	0.60a	1.98a	2.81a	1.68a	1.80b	2.31a	0.75b	2.74a	0.40b	
				**Row spacing 76 cm**						
	**Protein**	**Oil**	**C16:0**	**C18:0**	**C18:1**	**C18:2**	**C18:3**	**Sucrose**	**Raffinose**	**Stachyose**
**40K**	38.1b	23.88a	10.64a	3.62a	23.95a	56.12a	7.62a	3.51a	0.58a	4.03a
**56K**	38.7a	23.74a	10.35a	3.63a	24.54a	55.5a	7.86a	3.52a	0.57a	4.04a
	**ASP**	**THR**	**SER**	**GLU**	**PRO**	**GLY**	**ALA**	**CYS**	**VAL**	
**40K**	4.11b	1.895a	2.25a	4.63b	2.283b	2.604a	1.975a	0.275b	2.445a	
**56K**	4.14a	1.888a	2.25a	4.795a	2.296a	2.595a	1.975a	0.296a	2.441a	
	**MET**	**ISO**	**LEC**	**TYR**	**PHE**	**LYS**	**HIS**	**ARG**	**TRY**	
**40K**	0.604a	1.94b	2.74b	1.695b	1.795b	2.31a	0.779a	2.71b	0.44b	
**56K**	0.60b	1.97a	2.81a	1.721a	1.821a	2.31a	0.767a	2.77a	0.46a	

^a^ Seeding rate was used at 100,000 seeds/ac and 140,000 seeds/ac (i.e., 40,000 seeds ha^−1^ or 56,000 seeds ha^−1^ ) which is abbreviated as 40K and 56K throughout the manuscript, respectively. ^b^ Three configuration forms were used as follows: 15 inches twin row, 15 inch row spacing (40 cm) which is abbreviated as 15 row spacing, and 30 inch row spacing (76 cm) which is abbreviated as 30 row spacing. Values are means of four replications. Values followed by different lowercase letters within a column within each treatment are different at P ≤ 0.05.

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
