# Peer review of "The Influence of Agricultural Practices, the Environment, and Cultivar Differences on Soybean Seed Protein, Oil, Sugars, and Amino Acids"

_plants, 2020, doi:10.3390/plants9030378_

Round 1

Reviewer 1 Report

To authors,

This is a very interesting and well written paper. Data is analyzed with statistically adequate and useful parameters. Data is new and numerous.

I only have one main concern regarding how data is presented. Data is numerous and tables are too big and hard to read. I believe it needs to be rethink by the authors and could improve readability.

I will just give some ideas in how to do it, but it corresponds to the authors to decide how to do it:

In general, all tables could be resized to use less space

- Tables could show only the more significant parameters and the rest go to Supplementary material

- Anova parameters showing F statistics would be Supplementary material

- Table 1 to 4: two columns for F and P makes tables giant.

- Table 5: if 40K and 56K is used in the text, why not use the same in the tables to improve readability? Also, lines with aminoacid names could be in bold too.

- Table 5 and 6: could be 2015 and 2016 data in the same table? ex: showing data like this twin-row 40 cm Protein: 38.8a/39.3a

In general, all bar figures have too much empty space before and after the data is shown.

- Figure 1 and 2: Data on figure 1 and 2 (2015 and 2016) could be in the same figure one next to another. Also, then figure 1 and 2 could be merged in only one figure

- Figure 6. the sentence "bars without letter...." should be on the figure caption

- figures should all have the same codification, whether ns or no letter is used when no significative differences were found

Author Response

Dear Editor :

Please receive our revised manuscript entitled “Influences of agricultural practices, environment, and cultivar differences on soybean seed protein, oil, sugars, and amino acids” by Bellaloui et al. for the Special Issue “Physiological, Genetic, Molecular, and Environmental Factors Influencing Seed Nutrition”

We would like to thank the reviewers’ comments and appreciate their valuable comments. We believe that the reviewers’ comments improved the quality of our manuscript. We revised all comments of the reviewers (1 and 2) and responded to them with track changes as appropriate or as detailed below.

Our responses to reviewers’ comments are below:

Reviewer 1Comments and Suggestions for Authors

This is a very interesting and well written paper. Data is analyzed with statistically adequate and useful parameters. Data is new and numerous.

I only have one main concern regarding how data is presented. Data is numerous and tables are too big and hard to read. I believe it needs to be rethink by the authors and could improve readability.

I will just give some ideas in how to do it, but it corresponds to the authors to decide how to do it:

In general, all tables could be resized to use less space

- Tables could show only the more significant parameters and the rest go to Supplementary material

Response of authors: We deleted all columns of F values and kept only P values for Tables 1-4 for better clarity.

Reviewer Comment - Anova parameters showing F statistics would be Supplementary material

Response of authors: ANOVA tables are essentials, especially for breeders for breeding selection to see the effects of cultivar variation with other variables.

Reviewer Comment - Table 1 to 4: two columns for F and P makes tables giant.

Response of authors: We deleted all columns of F values and kept only P values for Tables 1-4 for better clarity

Reviewer Comment - Table 5: if 40K and 56K is used in the text, why not use the same in the tables to improve readability? Also, lines with aminoacid names could be in bold too.

Response of authors: Table 5: we are using now 40K and 56K as suggested in text. All amino acids are bold now.

Reviewer Comment - Table 5 and 6: could be 2015 and 2016 data in the same table? ex: showing data like this twin-row 40 cm Protein: 38.8a/39.3a

Response of authors: In Table 5, statistical analysis was done for 2015 only and comparison was done within each column for each variable. Adding 2016 horizontally next to 2015 will make the description in the results section more complicated as the comparison should be both vertically and horizontally as well, making the table explanation more complex. So, we left the tables 4 and 6 as they are, and we believe that the tables are clear and comprehensive.

Reviewer Comment: In general, all bar figures have too much empty space before and after the data is shown.

Response of authors: Figures are rearranged now and space was used more efficiency.

Reviewer Comment - Figure 1 and 2: Data on figure 1 and 2 (2015 and 2016) could be in the same figure one next to another. Also, then figure 1 and 2 could be merged in only one figure

Response of authors: Figure 1 and 2 are now in one page; Figure 3 and 4 are now in one page; Figure 5 and 6 are now in one page.

Reviewer Comment

- Figure 6. the sentence "bars without letter...." should be on the figure caption

- figures should all have the same codification, whether ns or no letter is used when no significative differences were found

Response of authors: The caption only explains the ns or no letter and it does not affect the consistency or the quality of figures at all. Adding ns to bars will make the bars crowded and will affect the quality of figures. Therefore, we suggest we keep them as they are as they are all explained and clarified in the caption.

Reviewer 2

Comments and Suggestions for Authors

This a complete manuscript with a complete description on the impact of practices on the composition of soybean seeds.

In the pdf you can find my comments and minor corrections.

The two main points that I want to be corrected is  first the relationship between oleic/linoleic/linolenic acid:

  • In the introduction lines 89-90 references cited are not appropriated as they don't discuss about this topic
  • Response of authors: We included the reference Xue et al. (2008) in the text and list of reference as suggested by the reviewer. We believe that the rest of references are important and related to the topic of the relationship between these fatty acids and the inverse relationship between oleic and linoleic and linolenic acids. This topic was discussed in the Discussion section and needs some references from previous research and what previous research found.
  • Reviewer Comment: In the results: L 128-129
  • Response of authors: We just stated what the results showed in the results section. We further explained the statement in the Discussion section for better clarity as shown in track-changes after L457, as you suggested in the next comment.
  • Reviewer Comment: In the discussion L457 oleic acid synthesis, so content, is controlled by temperature during grain filling, having no recognized role on environmental or chemical stressesResponse of authors: The literature indicated that although seed composition is genetically controlled, environment has significant effects on seed composition constituents including fatty acids oleic, linoleic, and linolenic. The main complexity is that very little is known to explain the relationships between the effects of environment and genetics on these fatty acids. We included in the text what the reviewer suggested in that temperature affects the fatty acid enzymes controlling fatty acids production. The objective of the current research is not to look at the yield, but it focuses on seed composition; so yield data is not available at the moment. Response of authors: The pdf file was reviewed and we addressed all comments on the pdf files as suggested above. The figures 1 and 2 were combined in one page as Figure 1; Figure 3 and 4 as Figure 2; Figure 5 and 6 as Figure 3. Bars are not necessary as letters representing the significance, resulted from statistical analysis by SAS, were used on bars (some articles published in Plants used the same approach and do not show bars, but used letters.
  • Reviewer Comment: The other points that can be shown in the pdf document are more form comments in the figures and tables and in the text. One suggestion is to combine figure 1 with the 4; figure 2 with 5; and figure 3 with 6. Also add error bars.
  • Second: About the impact of N fertilization L 333, the results are not so clear and we have no view on the yield. I would like that the results concerning the yield were shown, and a discussing is made about  the interest of fertilizing leguminous crops.

Reviewer 2 Report

This a complete manuscript with a complete description on the impact of practices on the composition of soybean seeds.

In the pdf you can find my comments and minor corrections.

The two main points that I want to be corrected is  first the relationship between oleic/linoleic/linolenic acid:

  • In the introduction lines 89-90 references cited are not appropriated as they don't discuss about this topic
  • In the results: L 128-129
  • In the discussion L457 oleic acid synthesis, so content, is controlled by temperature during grain filling, having no recognized role on environnemental or chemical stresses

Second: About the impact of N fertilization L 333, the results are not so clear and we have no view on the yield. I would like that the results concerning the yield were shown, and a discussing is made about  the interest of fertilizating leguminous crops.

The other points that can be shown in the pdf document are more form comments in the figures and tables and in the text. One suggestion is to combine figure 1 with the 4; figure 2 with 5; and figure 3 with 6. Also add error bars.

Author Response

Dear Editor :

Please receive our revised manuscript entitled “Influences of agricultural practices, environment, and cultivar differences on soybean seed protein, oil, sugars, and amino acids” by Bellaloui et al. for the Special Issue “Physiological, Genetic, Molecular, and Environmental Factors Influencing Seed Nutrition”

We would like to thank the reviewers’ comments and appreciate their valuable comments. We believe that the reviewers’ comments improved the quality of our manuscript. We revised all comments of the reviewers (1 and 2) and responded to them with track changes as appropriate or as detailed below.

Our responses to reviewers’ comments are below:

Reviewer 1 Comments and Suggestions for Authors

This is a very interesting and well written paper. Data is analyzed with statistically adequate and useful parameters. Data is new and numerous.

I only have one main concern regarding how data is presented. Data is numerous and tables are too big and hard to read. I believe it needs to be rethink by the authors and could improve readability.

I will just give some ideas in how to do it, but it corresponds to the authors to decide how to do it:

In general, all tables could be resized to use less space

- Tables could show only the more significant parameters and the rest go to Supplementary material

Response of authors: We deleted all columns of F values and kept only P values for Tables 1-4 for better clarity.

Reviewer Comment - Anova parameters showing F statistics would be Supplementary material

Response of authors: ANOVA tables are essentials, especially for breeders for breeding selection to see the effects of cultivar variation with other variables.

Reviewer Comment - Table 1 to 4: two columns for F and P makes tables giant.

Response of authors: We deleted all columns of F values and kept only P values for Tables 1-4 for better clarity

Reviewer Comment - Table 5: if 40K and 56K is used in the text, why not use the same in the tables to improve readability? Also, lines with aminoacid names could be in bold too.

Response of authors: Table 5: we are using now 40K and 56K as suggested in text. All amino acids are bold now.

Reviewer Comment - Table 5 and 6: could be 2015 and 2016 data in the same table? ex: showing data like this twin-row 40 cm Protein: 38.8a/39.3a

Response of authors: In Table 5, statistical analysis was done for 2015 only and comparison was done within each column for each variable. Adding 2016 horizontally next to 2015 will make the description in the results section more complicated as the comparison should be both vertically and horizontally as well, making the table explanation more complex. So, we left the tables 4 and 6 as they are, and we believe that the tables are clear and comprehensive.

Reviewer Comment: In general, all bar figures have too much empty space before and after the data is shown.

Response of authors: Figures are rearranged now and space was used more efficiency.

Reviewer Comment - Figure 1 and 2: Data on figure 1 and 2 (2015 and 2016) could be in the same figure one next to another. Also, then figure 1 and 2 could be merged in only one figure

Response of authors: Figure 1 and 2 are now in one page; Figure 3 and 4 are now in one page; Figure 5 and 6 are now in one page.

Reviewer Comment

- Figure 6. the sentence "bars without letter...." should be on the figure caption

- figures should all have the same codification, whether ns or no letter is used when no significative differences were found

Response of authors: The caption only explains the ns or no letter and it does not affect the consistency or the quality of figures at all. Adding ns to bars will make the bars crowded and will affect the quality of figures. Therefore, we suggest we keep them as they are as they are all explained and clarified in the caption.

Reviewer 2

Comments and Suggestions for Authors

This a complete manuscript with a complete description on the impact of practices on the composition of soybean seeds.

In the pdf you can find my comments and minor corrections.

The two main points that I want to be corrected is  first the relationship between oleic/linoleic/linolenic acid:

  • In the introduction lines 89-90 references cited are not appropriated as they don't discuss about this topic
  • Response of authors: We included the reference Xue et al. (2008) in the text and list of reference as suggested by the reviewer. We believe that the rest of references are important and related to the topic of the relationship between these fatty acids and the inverse relationship between oleic and linoleic and linolenic acids. This topic was discussed in the Discussion section and needs some references from previous research and what previous research found.
  • Reviewer Comment: In the results: L 128-129
  • Response of authors: We just stated what the results showed in the results section. We further explained the statement in the Discussion section for better clarity as shown in track-changes after L457, as you suggested in the next comment.
  • Reviewer Comment: In the discussion L457 oleic acid synthesis, so content, is controlled by temperature during grain filling, having no recognized role on environmental or chemical stressesResponse of authors: The literature indicated that although seed composition is genetically controlled, environment has significant effects on seed composition constituents including fatty acids oleic, linoleic, and linolenic. The main complexity is that very little is known to explain the relationships between the effects of environment and genetics on these fatty acids. We included in the text what the reviewer suggested in that temperature affects the fatty acid enzymes controlling fatty acids production. The objective of the current research is not to look at the yield, but it focuses on seed composition; so yield data is not available at the moment. Response of authors: The pdf file was reviewed and we addressed all comments on the pdf files as suggested above. The figures 1 and 2 were combined in one page as Figure 1; Figure 3 and 4 as Figure 2; Figure 5 and 6 as Figure 3. Bars are not necessary as letters representing the significance, resulted from statistical analysis by SAS, were used on bars (some articles published in Plants used the same approach and do not show bars, but used letters.
  • Reviewer Comment: The other points that can be shown in the pdf document are more form comments in the figures and tables and in the text. One suggestion is to combine figure 1 with the 4; figure 2 with 5; and figure 3 with 6. Also add error bars.
  • Second: About the impact of N fertilization L 333, the results are not so clear and we have no view on the yield. I would like that the results concerning the yield were shown, and a discussing is made about  the interest of fertilizing leguminous crops.